# Polymerization driven monomer passage through monolayer chemical vapour deposition graphene

Tao Zhang[1,2], Zhongquan Liao[3,4], Leonardo Medrano Sandonas [3,5], Arezoo Dianat[3], Xiaoling Liu[6], Peng Xiao[7], Ihsan Amin[1,8], Rafael Gutierrez[3], Tao Chen[7], Ehrenfried Zschech[2,4], Gianaurelio Cuniberti[2,3,9] & Rainer Jordan[1,2]

Mass transport through graphene is receiving increasing attention due to the potential for molecular sieving. Experimental studies are mostly limited to the translocation of protons, ions, and water molecules, and results for larger molecules through graphene are rare. Here, we perform controlled radical polymerization with surface-anchored self-assembled initiator monolayer in a monomer solution with single-layer graphene separating the initiator from the monomer. We demonstrate that neutral monomers are able to pass through the graphene (via native defects) and increase the graphene defects ratio (Raman $I_D/I_G$) from ca. 0.09 to 0.22. The translocations of anionic and cationic monomers through graphene are significantly slower due to chemical interactions of monomers with the graphene defects. Interestingly, if micropatterned initiator-monolayers are used, the translocations of anionic monomers apparently cut the graphene sheet into congruent microscopic structures. The varied inter-actions between monomers and graphene defects are further investigated by quantum molecular dynamics simulations.

[1] Chair of Macromolecular Chemistry, Faculty of Chemistry and Food Chemistry, School of Science, Technische Universität Dresden, Mommsenstr. 4, 01062 Dresden, Germany. [2] Center for Advancing Electronics Dresden, Technische Universität Dresden, 01062 Dresden, Germany. [3] Institute for Materials Science and Max Bergmann Center of Biomaterials, Technische Universität Dresden, 01062 Dresden, Germany. [4] Fraunhofer Institute for Ceramic Technologies and Systems (IKTS), Maria-Reiche-Straße 2, 01109 Dresden, Germany. [5] Max Planck Institute for the Physics of Complex Systems, 01187 Dresden, Germany. [6] Leibniz-Institut für Polymerforschung Dresden e.V., Hohe Straße 6, 01069 Dresden, Germany. [7] Key Laboratory of Marine Materials and Related Technologies, Zhejiang Key Laboratory of Marine Materials and Protective Technologies, Ningbo Institute of Material Technology and Engineering, Chinese Academy of Sciences, Ningbo 315201, China. [8] Junior Research Group Biosensing Surfaces, Leibniz Institute for Plasma Science and Technology, INP Greifswald e.V., Felix-Hausdorff-Strasse 2, 17489 Greifswald, Germany. [9] Dresden Center for Computational Materials Science, Technische Universität Dresden, 01062 Dresden, Germany. Correspondence and requests for materials should be addressed to T.Z. (email: tao.zhang@tu-dresden.de) or to T.C. (email: tao.chen@nimte.ac.cn) or to R.J. (email: rainer.jordan@tu-dresden.de)

Graphene and other two-dimensional materials are promising candidates for next-generation separation membranes owing to their atomic thickness[1–5]. In fact, a pristine graphene sheet is mostly impermeable to all atoms and molecules due to the unfavorable energy barriers of its closely spaced carbon atoms[6]. Even graphene prepared by chemical vapor deposition (CVD), which is expected to have Stone–Wales defects, is theoretically demonstrated to be impermeable to helium under ambient conditions[7]. As such, research has shown that graphene can be used as an effective barrier to oxidation of metal surfaces under certain conditions[8–10], as well as barrier for the deposition of self-assembled monolayers[11]. However, in another scenario, ions and molecules have been observed to be able to transport through single-layer graphene (via intrinsic defects) promoted by external pressure or concentration gradient[12–14]. More interestingly, Geim and colleagues[15] recently showed that, upon electronic potential, pristine graphene can be highly permeable to thermal protons under ambient conditions, and Geiger and colleagues[16] reported a reversible proton transfer through graphene from the aqueous phase silanol groups of a substrate surface, which correlate principally to the pioneer work of voltage-driven ionic transport across suspended CVD graphene by Golovchenko and colleagues[17] in 2010. These studies suggest that the permeability of single-layer graphene could be significant (e.g., overcome steric effect) if mass-transport is driven by an external force[18,19].

Intrigued by these experiments, we aim to address the following fundamental questions by taking advantage of surface-initiated polymerization (in which initiators can be selectively bonded to a surface to drive the movement and addition of monomers)[20]: Can possibly also organic molecules such as monomers pass through single-layer CVD graphene? If so, what happens to graphene? Is the size and/or the charge of the transported monomers of importance? Since atomic-thin graphene is transparent to van der Waals[21] and Coulomb forces[22], it should be possible that the graphene-covered radical initiator monolayer[23,24] can still interact with reactants (monomer, copper complex) on the other side of graphene. Meanwhile, the living nature of controlled radical polymerization (CRP) requires a continuous transport of the monomers through the graphene membrane and steadily consumes monomers near the graphene sheet to surface-bound polymer brushes (Fig. 1). Thus, the monomer concentration remains to be a step-function at any time of the surface-initiated controlled radical polymerization (SI-CRP)[20,25,26]. The morphology and thickness of the resulting polymer brush layer under the graphene are direct indicators if and how fast a respective monomer is translocated through the graphene.

In this work, we show that neutral monomers of various sizes are able to smoothly pass through monolayer CVD graphene under the driven force of radical initiator in polymerization, although the native defects of graphene are largely increased due to the translocation. In contrast, the passage of charged monomers is severely interrupted by the graphene. Notably, the translocation of anionic monomers selectively cuts graphene into congruent microscopic patterns. These results indicate that the translocation of a large molecule through monolayer CVD graphene can be realized by applying an external driven force to the molecule, and the charge of the molecule has a greater impact on the translocation than the molecule size.

## Results

**Sample preparation.** As shown in Fig. 2a, a micropatterned initiator-monolayer of 2-bromoisobutyryl bromide (BiBB) (Supplementary Figs. 1, 2) was prepared on a $SiO_2$ wafer piece[27] and covered by a monolayer of graphene (ca. $1 \times 0.5$ cm$^2$) from CVD. The typical surface morphology of CVD graphene with micrometer size wrinkles is revealed by atomic force microscopy (AFM,

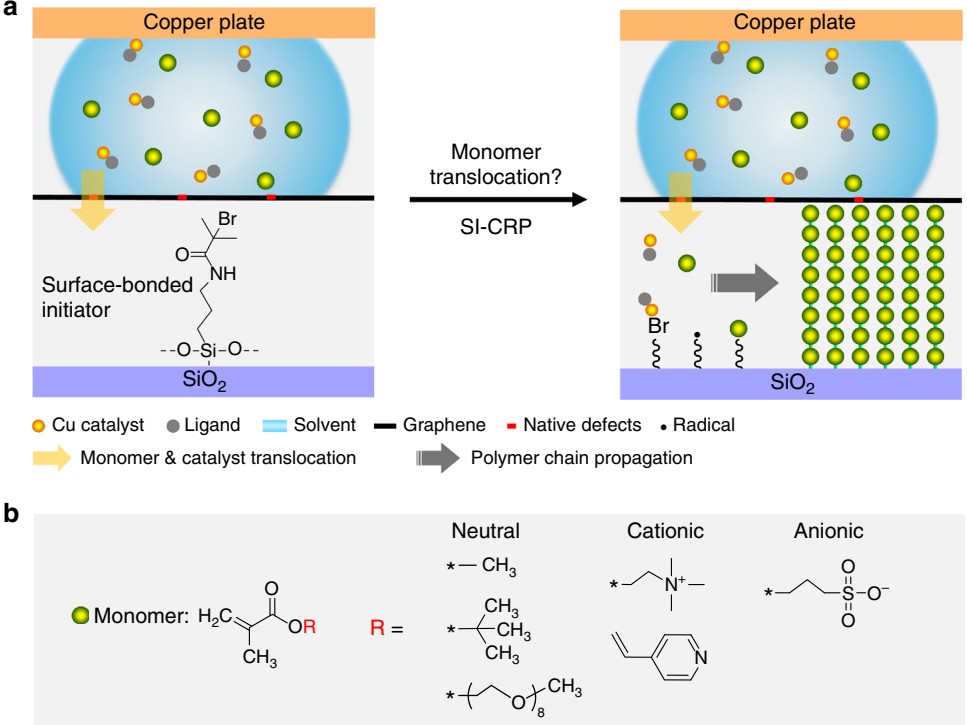

**Fig. 1** Passage of monomers through monolayer graphene driven by polymerization. **a** Schematic demonstration of the process of monomer translocation through monolayer graphene driven by surface-initiated controlled radical polymerization (SI-CRP). Surface-bonded polymer brushes can be formed on the $SiO_2$ surface after monomer translocation. **b** The molecular structures of different types of monomers (neutral, cationic, and ionic) investigated in this work

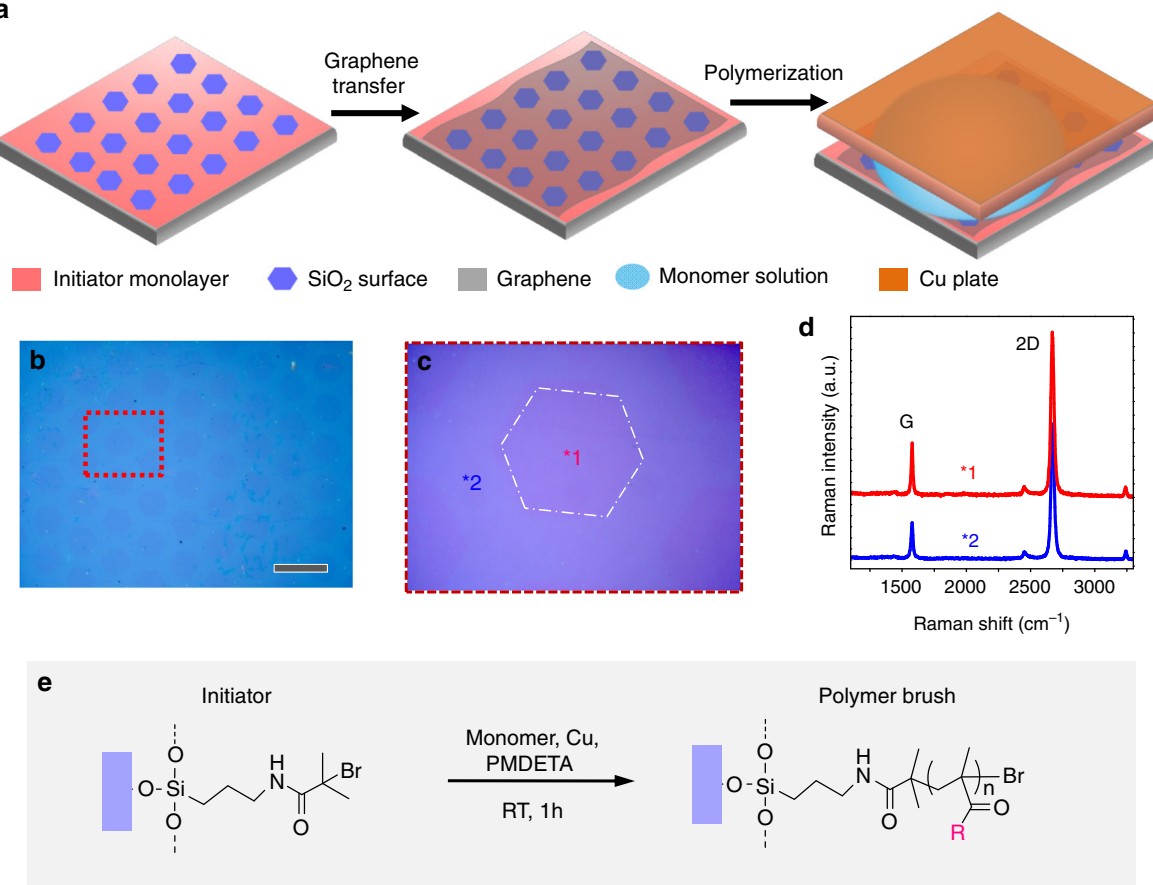

**Fig. 2** Substrate preparation and characterization. **a** Schematic illustration of the preparation of substrate graphene-BiBB-SiO$_2$ and SI-CRP. **b** Optical image of graphene-BiBB-SiO$_2$, scale bar: 40 μm. **c** A close-up of selected area from **b**. **d** Typical Raman spectra are given for the two selected positions from **c**. **e** Reaction scheme of SI-CRP on graphene-BiBB-SiO$_2$

Supplementary Fig. 3). The lattice structure of single-layer graphene was confirmed by selected area electron diffraction (SAED) pattern and high-resolution transmission electron microscope (HRTEM) (Supplementary Fig. 4). The complete coverage of the initiator-SAM area is visible by optical microscopy and initiator-SAM patterning is still visible (Fig. 1b, c). The high quality of graphene after the transfer was verified by Raman spectroscopy (Fig. 2d). Raman intensity ratio ($I_D/I_G$) mapping results suggest that the whole graphene sheet is rather homogeneous with low defect concentration ($I_D/I_G < 0.1$, Supplementary Fig. 5). The graphene-covered initiator-SAM was then faced to a Cu plate as catalyst source in a distance of ca. 0.5 mm. The gap between initiator-SAM and Cu plate was filled by a polymerization solution of dimethyl sulfoxide (DMSO) or water[24,28,29] containing the monomer (MMA, METAC, or SPMA), the ligand (1,1,4,7,7-pentamethyldiethylenetriamine, PMDETA) for polymerization (Fig. 2e). After 1 h reaction at room temperature, the Cu plate was removed and the substrate was thoroughly cleaned (see experimental) to remove all traces of physisorbed monomer or polymers and investigated.

**Neutral monomers: unimpeded translocation through graphene.** Despite the graphene barrier, the SI-CRP of methyl methacrylate (MMA) readily gave poly(methyl methacrylate) (PMMA) brush layers as shown in Fig. 3a. The AFM topographic scan gives a surprisingly thick PMMA brush with a brush growth rate of 79 nm h$^{-1}$ being only slightly slower as compared to the same SI-CRP at uncovered initiator-SAMs (88 nm h$^{-1}$) (Fig. 3b and

Supplementary Fig. 6). The formation of PMMA brushes and also the difference of graphene covered and uncovered areas can be readily seen by optical microscopy (Fig. 4a). Interestingly, the SI-CRP continues for the entire reaction time of 1 h which requires a steady flow of MMA through the graphene barrier. In a close-up visualization by scanning electron microscope (SEM), randomly-distributed polymer humps were observed on the graphene surface (Supplementary Fig. 7), which are inferred to be of the preferred path (e.g., graphene grain boundary) for the monomer transport. The surrounded flat regions (with sizes from hundreds of nm to a few μm) reveal the grain lattice of CVD graphene, in agreement with a typical grain size of CVD graphene (ranging from 250 nm to 3–5 μm) grown on Cu foil[30,31]. Similar results were obtained with other neutral monomers such as tert-butyl methacrylate (tBuMA, 3.9 × 5.9 Å, Supplementary Fig. 8a) and even much bulkier oligo(ethylene glycol) methyl ether methacrylate (OEGMA$_{475}$, 3.8 × 35.2 Å, Supplementary Fig. 8b).

The graphene coverage was fully retained at both regions with (*1) and without (*2) initiator-SAM/polymer brush underneath, as confirmed by AFM (Fig. 3b and Supplementary Fig. 9) and Raman spectroscopy (Fig. 3c). Detailed Raman mapping (Fig. 3d–f and Supplementary Fig. 10) revealed several interesting changes (in region *2) on graphene due to the translocation of MMA. Firstly, the $I_D/I_G$ intensity ratio (i.e., defects concentration) increased from ca. 0.09 to ca. 0.22 (Fig. 3e), which suggests that the passage of MMA through graphene greatly enlarged native defects of the lattice, since the size of graphene boundary defects (ca. 2 Å)[16] are normally much smaller than that of MMA

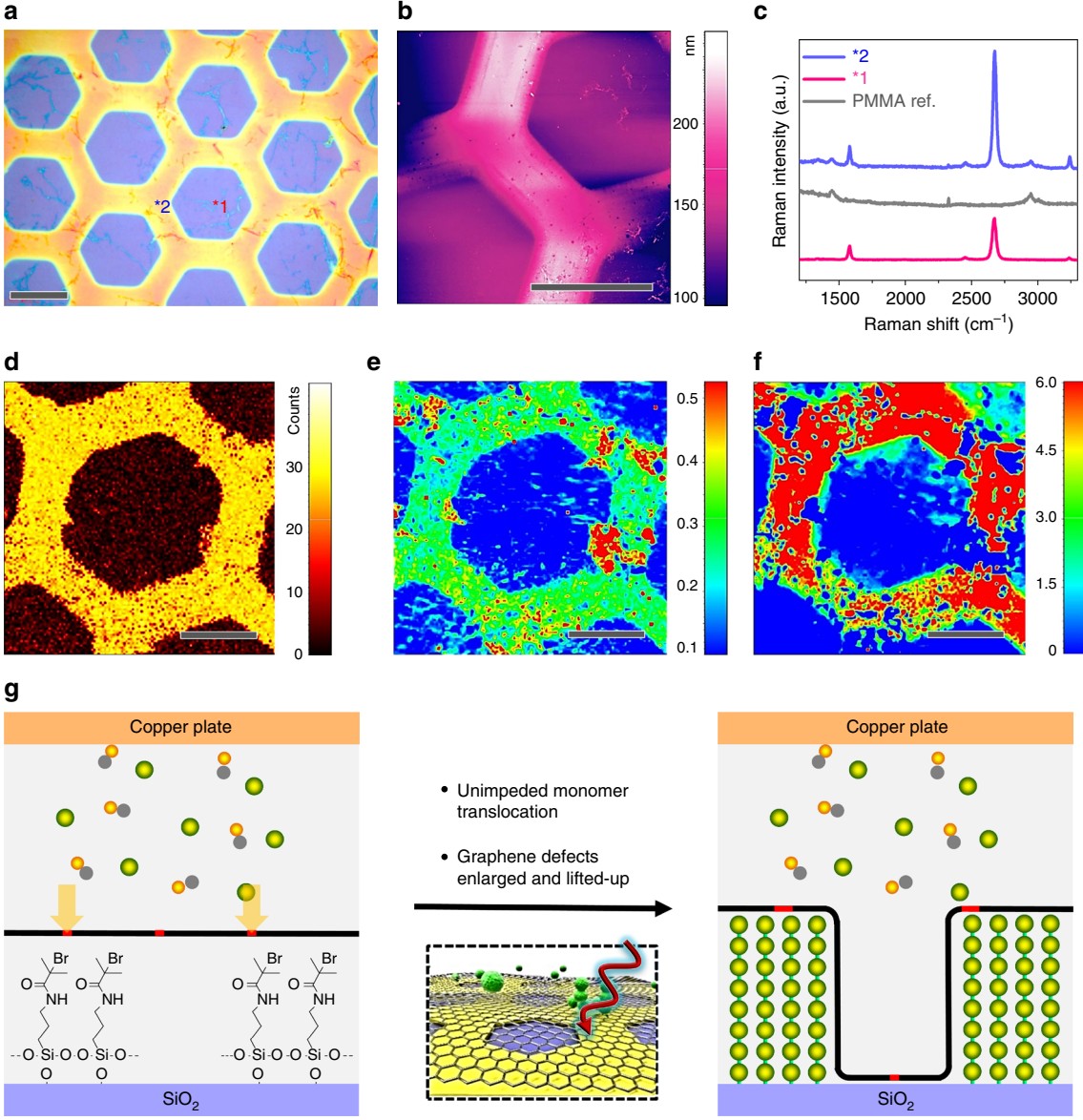

**Fig. 3** Translocation of MMA through graphene driven by SI-CRP. **a** Optical micrograph and **b** AFM topographic scan of PMMA grafted on graphene-BiBB-SiO$_2$. **c** Representative Raman spectra of regions *1 and *2 as marked in **a**. **d** Raman mapping with the integrated intensity at 2950 cm$^{-1}$ (PMMA). **e** Raman mapping of the integrated intensities ratio of $I_D/I_G$ (graphene). **f** Raman mapping of integrated intensities ratio of $I_{2D}/I_G$ (graphene). **g** Scheme of the experimental set-up and the translocation of MMA through graphene induced by SI-CRP. Scale bars in **a**, **b**: 40 μm; in **d–f**: 20 μm

(3.8 × 6.6 Å, Supplementary Fig. 11). In addition, the ratio of $I_{2D}/I_G$ was improved from ca. 2.8 to 7.0 (Fig. 3f). Similar $I_{2D}/I_G$ has been observed on suspended single-layer graphene[32–34], implying the graphene (of region *2) was lifted from the substrate by the grown PMMA brush underneath.

Furthermore, we found that the characteristic SAED patterns of graphene are still present after monomer translocation (Supplementary Fig. 12), but their intensity becomes much weaker than original graphene at the same characterization conditions. In addition, faint ring-like SAED patterns (Supplementary Fig. 12c, d) are observable in selected regions rather than six-spots of pristine graphene. The decrease of SAED intensity is due to the increase of defect concentration of graphene after the translocation of monomer, which is continuous with Raman results. The generation of ring-like SAED pattern is probably because of the distortion of graphene lattice since the translocation/polymerization preferably occurred at native defects.

Unfortunately, the newly generated defects/pores could not be viewed by HRTEM due to the interference of PMMA brush layer (Supplementary Fig. 13).

These experiments show that neutral monomers even as big as tBuMA or OEGMA$_{475}$ can pass through single-layer graphene and the polymer brush lifts the graphene sheet from the substrate. It is noteworthy that analog experiments carried out with two layers of CVD graphene sheets covering the initiator-SAM did not show indications of polymer brush formation (Supplementary Figs. 14, 15). This also indicates that the monomer is in fact translocated through the graphene single-layer and not diffusing from the graphene sheet edges in the space between the substrate and the graphene sheet. The control results, together with the evidence of defects ratio increase ($I_D/I_G$ and SAED, Fig. 3e and Supplementary Fig. 12) strictly along the initiator pattern, demonstrate the monomer pass through graphene via native subnanometer defects rather cracks or sheet edges.

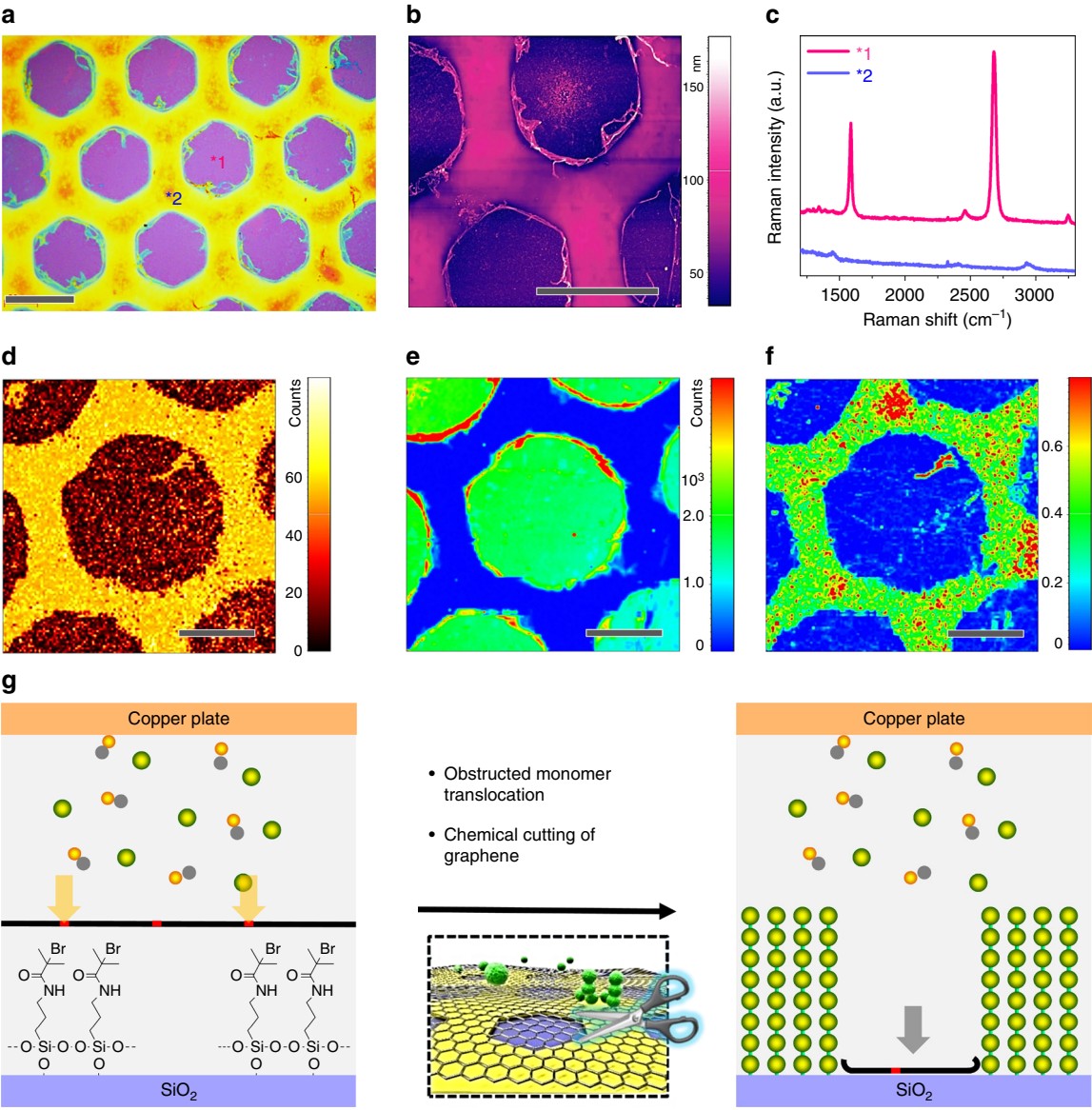

**Fig. 4** Translocation of SPMA through graphene driven by SI-CRP. **a** Optical microscopic image. **b** AFM topographic image. **c** Representative Raman spectra of regions *1 and *2 marked in **b**. **d** Raman mapping with the integrated intensity from PSPMA in the range of 2927 cm$^{-1}$. **e** Intensity Raman mapping of graphene 2D band. **f** Raman mapping of integrated intensity ratio of $I_D/I_G$ of graphene. **g** Scheme of the experimental set-up and the chemical cutting of graphene by the translocation of SPMA induced by SI-CRP. Scale bars in **a**, **b**: 40 μm; in **d**–**f**: 20 μm

**Anionic monomer: chemical cutting of graphene.** Secondly, an anionic monomer, 3-sulfopropyl methacrylate potassium salt (SPMA) was converted as described above. Again, the SI-CRP of SPMA resulted in PSPMA brushes but apparently the graphene was cut into hexagons congruent to the initiator-SAM pattern (Fig. 4a–c, Supplementary Figs. 16, 17). Raman intensity mapping at 2927 cm$^{-1}$ verified that a PSPMA brush was selected grafted at the pattern initiator-SAM areas (*2, Fig. 4d and Supplementary Fig. 18). Raman spectra and mapping of $I_{2D}$ of graphene showed that the graphene was retained only in the region without initiator-SAMs (*1) (Fig. 4c, e). And the defect density (i.e., $I_D/I_G$) of graphene at this region (*1) was unchanged (Fig. 4f and Supplementary Fig. 5), suggesting the polymerization solution itself has no detectable effects on graphene. In contrast to the experiments with neutral MMA, the growth rate of the respective brush for SPMA was significantly decreased by ca. 65% (Supplementary Table 1). In graphene covered areas, the brush growth rate was 43 nm h$^{-1}$ as compared to uncovered initiator-SAMs

(121 nm h$^{-1}$) (Supplementary Fig. 19). We would like to note that the cutting of graphene layer was already started, even the thickness of PSPMA brush layer is only 8–10 nm (Supplementary Fig. 20). Such thin layer PSPMA brush below graphene has neglectable stretching force on graphene lattice. Therefore, we can conclude that the cutting of graphene by SPMA is due to the chemical interactions between SPMA and graphene defects, rather mechanical stretching/tearing.

Finally, a cationic monomer ((methacryloyloxy)ethyl trimethylammonium chloride, METAC) was investigated. As for the anionic monomer, also the transport of METAC through the graphene was significantly slower as revealed by the slower polymer brush growth rate (decreased by 60% from 266 nm h$^{-1}$ (without graphene) to 100 nm h$^{-1}$ (Supplementary Fig. 21 and Supplementary Table 1)). However, in contrast to the experiments with anionic monomers, the graphene was not cut by the monomer pass-through, and in contrast to the neutral monomers, island-like polymer plateaus varying from nanometers to

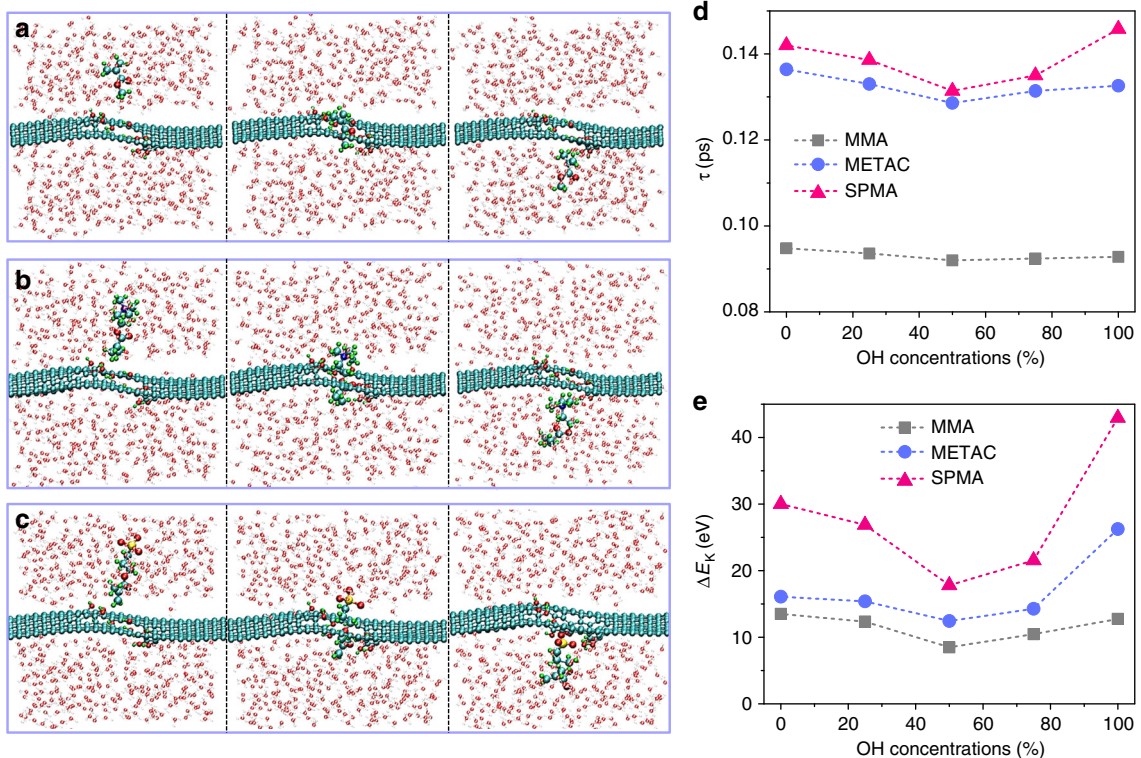

**Fig. 5** Simulation of monomers translocation through graphene defects. The snapshots of different monomers translocation through a graphene defect passivated with OH termination, **a** MMA, **b** METAC and **c** SPMA. Variation of the **d** translocation time $\tau$ and **e** the kinetic energy change $\Delta E_K$ as a function of the OH concentration for the monomers in the initially $CH_3$-passivated nanopore. $\Delta E_K = E_{K,bef} - E_{K,aft}$, with $E_{K,bef}$ and $E_{K,aft}$ as the kinetic energy of the monomer before and after the translocation process, respectively

micrometer were observed upon PMETAC brush formation, probably due to strong graphene–cation interaction at graphene defects sites[17]. A detailed analysis of the samples by confocal Raman mapping is presented in Supplementary Fig. 22.

Similar translocation behavior was observed for another cationic monomer (4-vinylpyridine (4VP), Supplementary Fig. 23). Dangling bonds or functional groups of the graphene defect sites interact with the passing monomers, and resulted in different passage behaviors through the graphene. As the size appears to be of minor importance in the range investigated (see results on neutral monomers), the charge of the passing molecules appears to have a much stronger influence.

Quantum molecular dynamics simulations were performed with a density functional-based tight-binding (DFTB) approach[35], to study the kinetic energy change $\Delta E_K$ during the translocation process through the graphene nanopore of various edge terminations for the three monomers SPMA, MMA, and METAC (see Fig. 5 and Supplementary Figs. 24, 25). We found that in all cases the translocation process for neutral MMA is much faster and easier than charged ones, as evidenced by the lowest values of translocation time $\tau$ and the loss of kinetic energy $\Delta E_K$ (Fig. 5d, e). In addition, the translocation of MMA is nearly independent to the OH concentration at the graphene edges. However, the translocation of charged monomers is obviously affected by the concentrations of OH edges. This is due to the stronger interaction between charged monomers (METAC and SPMA) and the graphene edge groups, which also enlarges the fluctuations of graphene defects dimension (Supplementary Fig. 26). For the fully OH-passivated pore, the highest loss of kinetic energy $\Delta E_K$ ($E_{K,bef} - E_{K,aft}$, with $E_{K,bef}$ and $E_{K,aft}$ being the kinetic energy of the monomer before and after the translocation process, respectively) is found for the negatively charged SPMA

(Supplementary Figs 25a), followed by the positively charged METAC and the neutral MMA (Supplementary Movies 1–3). Consequently, the fastest translocation time ($\tau \approx 0.093$ ps) has been found for neutral MMA. To quantify the degree of distortion of the nanopore during the translocation process, the change in the pore size ($D_H$) was calculated by computing the distance between diametrically opposed OH groups in the functionalized pore at each simulation time step, and then performing an average over the calculated distances (Supplementary Fig. 25b). The smallest degree of edge distortion was found for MMA ($D_H \approx 7.94$–7.08 Å), followed by METAC ($D_H \approx$ 7.86–8.04 Å) and SPMA ($D_H \approx 7.92$–8.24 Å). In the case of ether termination[10], both quantities, $\tau$ and $\Delta E_K$, smoothly decrease upon increasing the ether concentration, since the hole diameter gets larger and, hence, the monomers have less interaction with graphene defect edge to pass through (Supplementary Fig. 27).

The combined experimental and simulation results indicate that electrostatic interactions between charged species (monomers) and the functionalized graphene nanodefects are dominating the translocation process. Moreover, the much higher energy loss experienced by the charged monomers may explain (i) the strong obstruction effect observed in the translocation of both cationic and anionic monomers, and (ii) the observed chemical cutting (in the case of anionic monomers), since the larger dissipated energy into graphene may lead to a larger degree of structural distortion of the nanodefects.

## Discussion
We suggest that there are two possible passways for monomers transport at the graphene interface during SI-CRP (Fig. 6): (i) Vertical translocation through graphene native defects[36–38], which either enlarges the defects or completely breaks graphene

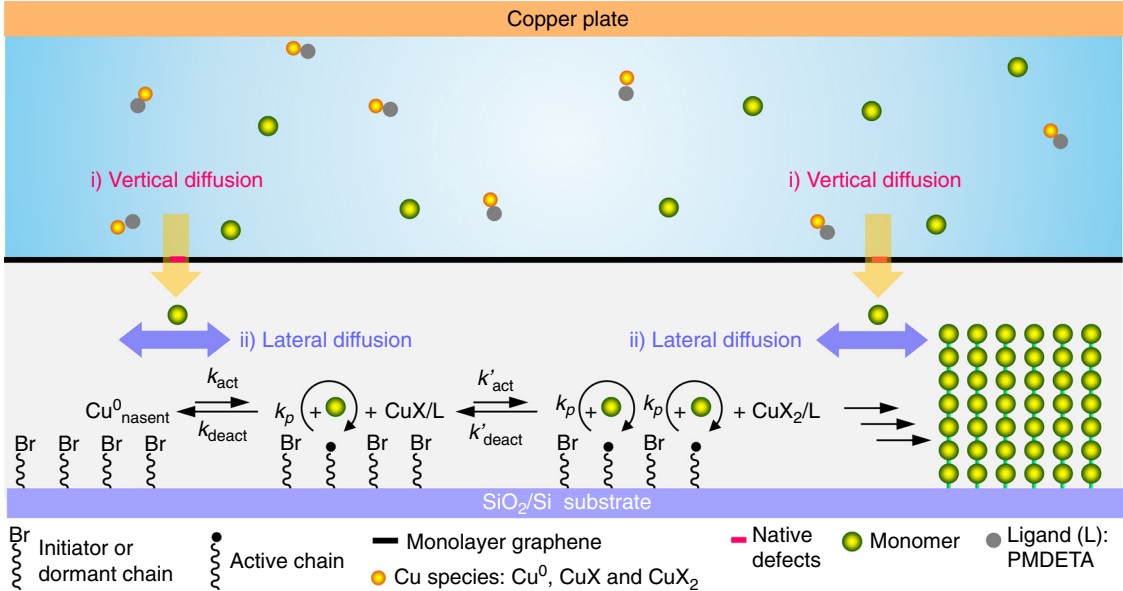

**Fig. 6** Schematic description for the translocation of monomers at a graphene interface. Vertical translocation of monomers occurs at graphene native defects. After passing through the graphene, the monomers diffuse laterally under graphene lattice

depending on the charges of monomers. (ii) After vertical translocation, the monomers are able to transfer laterally underneath of graphene lattice, which accumulates as polymer brush under graphene and gradually delaminates graphene from substrate[24]. Both of these two processes are ensured by the atomic thinness and flexibility of graphene monolayer[21] and strong Coulomb interactions between monomer and surface-bonded initiator in SI-CRP.

In conclusion, we found that large molecules such as vinyl monomers of different sizes can pass through single-layer CVD graphene if driven by a proximal chemical reaction such as SI-CRP. Furthermore, the SI-CRP continues and the formation of polymer brushes indicate a constant flow of monomers through the graphene. Notably, we found that the transport process is more charge-selective than size-selective. For example, a neutral monomer could smoothly translocate to the other side of graphene, by enlarging the graphene native defects. However, the translocation of cationic and anionic monomers was significantly obstructed (60 and 65% in polymerization rates), due to the strong interactions between charged monomers and graphene defects. These results on the translocation of actual molecules through a single layer of a two-dimensional material are intended to trigger further studies on the transport phenomena through this exciting class of materials. An additional aspect of this work is the possibility of the chemical cutting with techniques used for nano- and micropatterned polymer brushes[39] of two-dimensional materials into desired patterns or shapes for heterostructures[40].

## Methods

**Materials**. MMA, *t*BuMA, OEGMA475, and 4VP were purchased from Sigma-Aldrich (Weinheim, Germany), purified before use by passing through a basic alumina column to remove the inhibitor. 3-SPMA, METAC, 2-BiBB, 3-aminopropyltriethoxysilane (APTES), PMDETA (99%), dichloromethane (DCM, dry), DMSO, triethylamine (TEA), acetone (dry), and methanol (all from Sigma-Aldrich) were used as received. For reactions and water contact angle measurements, bidistilled deionized water was used. Cu plate (MicroChemicals GmbH, Germany): 1-side polished, p-type (boron), total-thickness-variation (TTV) < 10 μm, 1–10 Ohm cm; 10 nm Ti adhesion layer; 200 nm Cu (purity >99.9%), root-mean-square roughness <10 nm. The copper plate was consecutively washed with portions of 3 M HCl (in methanol), methanol and ethanol under ultrasonication (2 min), and dried under a flow of argon. The cleaned copper plate was immediately used for reactions.

**BiBB-functionalized SiO₂**. Silicon wafer pieces with a 300 nm oxide layer were obtained from Wacker AG, Burghausen, Germany. To remove any grease or other contaminants, silicon substrates were cleaned with piranha solution ($H_2O_2$:$H_2SO_4$, 1:3 v/v, 90 °C, 45 min; WARNING: Piranha solution reacts violently with organic matter!), washed extensively with bidistilled water, and dried with a stream of dry argon.

The silicon wafer substrate was then amine-functionalized by immersing in a 5% (v/v) APTES solution in dry acetone and subjected to treatment with ultrasound for 45 min. After SAM formation, the samples were extensively rinsed with dry acetone and dried under argon atmosphere. The substrate was then immersed in dry DCM (20 mL) under a nitrogen atmosphere. TEA (0.4 mL) was added, followed by dropwise adding initiator BiBB (20 mL, 2% in DCM) at 0 °C, and the reaction solution was allowed to stand for 24 h with stirring at 20 °C. The substrate was removed, washed with DCM, water, ethanol, and acetone, and then dried under a nitrogen stream. The quality of this surface-bonded initiator preparation method was confirmed by X-ray photoelectron spectroscopy (XPS) and water contact angle measurements. The total surface layer thickness (i.e., the native silicon dioxide layer plus the initiator) was ca. 2.0 nm, as judged by ellipsometry.

Patterned initiator was achieved on a selectively etching of as-prepared uniform initiator on $SiO_2$ by UV illumination (200 W Hg (Xe) lamps, LOT-oriel, Germany) through a photomask. The samples were clamped with Cu TEM grids and irradiated for 45 min at a distance of ca. 10 cm.

**Preparation of graphene-BiBB-SiO₂**. To transfer graphene to the initiator (BiBB) modified $SiO_2$, one side of the single-layer graphene (graphene supermarket) was coated with PMMA resist (Allresist GmbH product no. AR-P671.04, dissolved in chlorobenzene) and cured at 90 °C for 10 min. The other side of the sample was exposed to $O_2$ plasma to remove the graphene on that side. The Cu substrate was etched away by an aqueous solution of ammonium persulfate (0.25 g mL⁻¹) over a period of 2 h. After being rinsed thoroughly with deionized water, the PMMA-graphene film was transferred to a target substrate. The samples were naturally dried in air for 1 h and stored in high vacuum at room temperature for 24 h to enhance the adhesion of graphene with BiBB-SiO₂ surface. PMMA was removed by thorough rinsing in acetone and cured in isopropyl alcohol.

**SI-CRP on graphene-BiBB-SiO₂**. Polymerization solution preparation: (i) MMA: 1 mL monomer, 0.5 mL DMSO, 18.4 μL PMDETA; (ii) METAC: 1 mL monomer, 1 mL $H_2O$ and 0.5 mL methanol, 18.4 μL PMDETA; (iii) SPMA: 1 g monomer, 1 mL $H_2O$ and 0.5 mL methanol, 18.4 μL PMDETA.

The graphene-BiBB-SiO₂ was sandwiched with a copper plate (MicroChemicals GmbH, Germany) at a distance of $D = 0.5$ mm adjusted by two spacers[24,29]. A drop (ca. 20 μL) of monomer solution prepared above was introduced to the confined space between copper plate and graphene-BiBB-SiO₂. The assembly was left for typically 1 h at room temperature. After reaction, the plate was separated and the substrate immediately washed with either fresh DMSO or methanol and water mixture. Finally, the substrates were exhaustively rinsed with acetone to remove all traces of monomer solution, and subsequently dried in a stream of nitrogen.

**Molecular dynamics simulations.** The simulations of the monomers translocation through graphene defects were carried out by means of a DFTB approach using the DFTB+ code. This method combines accuracy with numerical efficiency and allows to efficiently deal with up to 2000 atoms in a quantum simulation. We have used the Slater–Koster parameters developed by Niehaus et al.[41] for the C, H, O, N, and S atoms. In all simulations, explicit water has been considered as parameterized in DFTB+ code, in which the interactions among water molecules are given by dipole–dipole as well as van der Waals interaction. In order to take into account van der Waals forces, dispersion corrections were included via Lennard–Jones potentials. The geometry optimization for the monomers and edge-terminated graphene was performed until the absolute value of the inter-atomic force lies below $10^{-4}$ atomic units. Periodic boundary conditions in all supercell directions were applied. We first generated a graphene pore defect with $CH_3$ termination and, then, the influence of OH termination concentration is studied. The influence of ether termination on the monomers translocation process through an initially OH-passivated pore has also been analyzed (see Supplementary Fig. 27). The dimension of the hole only allows us to consider 12 functional groups for each passivation state. Geometry optimization provided the most stable edge conformations in each case (Supplementary Fig. 24). The monomers were then placed approximately 10 Å on top of the graphene surface. The simulation supercells were then filled with water molecules. The left panels of Fig. 5a–c show the initial conformations of the different monomer/water/graphene systems. Microcanonical MD simulations were then run with an initial monomer velocity of roughly 100 Å $ps^{-1}$.

**Characterizations.** AFM scans were recorded with an Ntegra Aura (NT-MDT) atomic force microscope with a SMENA head in the semicontact mode. The used probes have a typical curvature radius of 6 nm, a resonant frequency of 47–150 kHz, and a force constant of 0.35–6.10 N $m^{-1}$. Raman spectra and maps were measured on a NT-MDT confocal spectrometer with a 532 nm laser, and the spot size of the laser beam was ca. 0.5 μm. The step size of Raman spatial mapping was ca. 0.5 μm, and the spectral resolution was 3 $cm^{-1}$ (obtained with a 600-grooves per mm grating). The Si peak at 520 $cm^{-1}$ was used as a reference for wavenumber calibration. The peaks were fitted with a single Lorentzian line shape to determine peak position and full width at half maximum. The optical microscopy images were measured on Zeiss Axioskop 2 MAT microscope.

## Data availability

The data that support the findings of this study are available from the corresponding authors on request.

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

## Acknowledgements

T.Z., I.A. and R.J. acknowledge financial support by the Cluster of Excellence "Center for Advancing Electronics Dresden" (cfAED). L.M.S. thanks the International Max Planck Research School Dynamical Processes in Atoms, Molecules and Solids and the Deutscher Akademischer Austauschdienst (DAAD) for the financial support. T.C. thanks the Natural Science Foundation of China (51573203) and the Key Research Program of Frontier Sciences at the Chinese Academy of Sciences (QYZDB-SSW-SLH036). We

acknowledge the Center for Information Services and High Performance Computing (ZIH) at TU Dresden for computational resources. We acknowledge support by the Open Access Publishing Funds of the SLUB/TU Dresden.

## Author contributions

T.Z. and R.J. conceived and designed the experiments and wrote the paper. T.Z. carried out most of the experiments. Z.L. and E.Z. assisted with the TEM characterization. L.M. S., A.D., R.G. and G.C. carried out molecular dynamics simulations and related data analysis. I.A. and X.L. assisted in sample preparation. P.X. and T.C. contributed to the discussions and enhancement of the manuscript. All authors discussed the results and commented on the manuscript.

## Additional information

**Competing interests:** The authors declare no competing interests.

