## [Peer Review File · Nature Communications]

Reviewers' comments:

Reviewer #1 (Remarks to the Author):

This is an interesting manuscript describing PCP through defects in single layer graphene. The experiments are of good quality and generally support the model and conclusions. A few questions should be answered:

- 1) What is the ionic strength in the experiments with water?
- 2) How are the water and counter ions treated in the DFT-B simulations?
- 3) The OH-passivated defects are mentioned, as far as this reviewer can tell, for the first time at the bottom of page 8 - are other defect terminations important as well (viz. Fig. S12 of ref. 10)? The brief text on the MD simulation details seems to indicate thought has been given to this topic - perhaps the authors can expand on that thought? Would differences in termination open the possibility to gate the process somehow?

As "very" is not a quantifiable term, its numerous appearances in the main text should be avoided.

Overall, an intriguing finding that should be reviewed again before acceptance.

Reviewer #2 (Remarks to the Author):

I was a pleasure to read this manuscript. It is well written, thoughtful and poses an interesting research question. The authors have also been very rigorous in the design and execution of the experiments that they report in this paper.

That being said, I have a number of thoughts about the question the authors are asking as well as the insight that this study provides and how much of interest and importance the work is for the broader field.

The question that motivates the work in this paper is whether molecules can pass through a single-layer of graphene. This is also reflected in the title, which mentions "translocation through graphene" as well as in the conclusions, where the authors write that "we found that ... monomers can diffuse through a single layer of graphene". However, what the data in the paper really show (and which the authors also mention in the discussion) is that monomer transport does not take place strictly through graphene, but instead takes place through defects / grain boundaries. These are two very different things ! Transport does not take place through graphene, but is related to the defects and microstructure of the CVD deposited graphene film. As a consequence, this paper does not teach the reader about graphene, but rather about the effects of defects !

The authors also propose that the findings of their work could be put to use for chemical cutting of 2D materials. If the observations that are reported in this paper, however, are not due to the chemical structure and properties of graphene, but are due to defects, I wonder how one can take advantage of this in a predictable manner.

In summary, I think this is a great set of experiments which are worth publication, but I think the work is more suited for a more specialized nano/graphene oriented journal.

Reviewer #3 (Remarks to the Author):

This manuscript presents very interesting observations of surface polymerization in a system consisting of a surface-grafted pattern of initiators, coated with a single layer of CVD graphene.

Different kinds of monomers (neutral, charged, different molecular weights) are introduced on the opposite side of the graphene, and the ensuing polymerization is examined by AFM and optical microscopy. The materials are also thoroughly characterized by a number of other techniques, including Raman, XPS, HRTEM. The authors find that polymer brushes grow despite of the presence of the graphene coating. Raman spectroscopy suggests that the polymer brushes form below the graphene, since the I_{2d}/I_g ratio increases after polymerization. Interestingly, the authors find that the polymer brush growth rate is almost the same as that in the absence of graphene for neutral monomers, but is reduced in the case of charged monomers. The authors also observe that polymerization with an anionic monomer 'cuts' the graphene, such that the polymer brush is no longer covered by graphene. To understand the observations, MD simulations are performed. The authors conclude that neutral molecules pass unimpeded through graphene, and that cutting of graphene is due to the interaction of the charged monomers with defects in graphene.

Overall, the experimental work is thorough and very well-executed. To the reviewer, it seems that the key conceptual novelty of this work is in the performance of a surface-initiated polymerization reaction on a graphene-coated surface, showing that it is possible to polymerize molecules in patterns below a graphene-coated surface. The work has some significant drawbacks, though. First, the manuscript is written in a highly speculative fashion, and makes several assertions that seem quite erroneous or not supported by the data. Second, the manuscript does not reflect much understanding of mass transport, but proceeds to make either incorrect or somewhat inaccurate statements about it, as seen in specific comments below. Third, the manuscript has errors and lacks clarity in some places, as outlined below. The reviewer believes that, if these issues are adequately addressed, the manuscript could be of suitable quality to be publishable in Nature Communications.

There is a significant amount of literature on defects in CVD graphene and mass transport across single-layer CVD graphene that is directly pertinent to this work (e.g., DOI: 10.1016/j.carbon.2015.12.089). This literature is in the context of graphene synthesis/characterization, corrosion resistance imparted by graphene coatings (e.g., DOI: 10.1021/jacs.5b08729), single-layer graphene membranes, and some other papers, e.g., graphene has been shown to be a barrier for self-assembly of SAMs (DOI: 10.1021/acsnano.5b03936). This literature needs to be reviewed and the manuscript needs to be revised accordingly. For example, the abstract states that 'no results for larger molecules (than protons, ions, water) through pristine graphene were reported', but that is not quite true (e.g., DOI: 10.1002/adma.201605896).

Page 2, line 4 from top: It is stated that literature suggests that CVD graphene is impermeable. However, there are several papers on transport of molecules across CVD graphene. While there is consensus that pristine graphene is impermeable to gases, molecules, and ions (except protons), it is also widely acknowledged in the literature that CVD graphene has defects that allow these to pass through.

Page 3 and SI Fig. 3: It is suggested that graphene grains are visible in the AFM image in SI Fig. 3. However, they are much more likely to be wrinkles than grains.

Page 4, top: Acronyms need to be explained. The manuscript also uses too many acronyms that makes it hard to read.

Page 4, line 3 from top: It is stated that the Cu plate was removed, but the Cu plate has not been mentioned before.

Page 4, bottom line: The sentence 'Apparently, passage of a large molecule such as MMA through a single-layer graphene can be realized by a graphene-proximal reaction such as CRP' is somewhat misleading. The reaction is not necessary for transport of the molecules across graphene; it merely

facilitates transport by capturing any molecules that cross over and in effect establishing a concentration gradient across graphene.

Page 5: 'This suggests that the passage of MMA through graphene greatly enlarged native defects' – There are several reasons why the I2d/Ig ratio could decrease, including contamination, change of support, or polymerization stress-induced damage to graphene. There is no indication to suggest that native defects are being enlarged.

Page 5, line 4 from bottom: For 2-layer graphene, AFM imaging should be performed to conclude that there is no polymerization. SI Fig. 13 only shows an optical image.

Page 6 and top of page 10, 'chemical cutting of graphene': It seems to the reviewer that the most plausible mechanism for 'chemical cutting' of graphene is the reduced adhesion to the anionic polymer brush that forms below graphene. Since the patterns are at the 10 micron scale, the graphene will delaminate, generating the pattern. The reason why neutral or cationic monomers do not 'cut' the graphene may be due to the stronger adhesion with graphene (hydrophobic, pi-pi, and cation-defect). It is not clear to the reviewer how the conclusion about chemical cutting can be drawn from the simulations.

Page 8-9, simulations: It is not clear how the simulations can be interpreted. The molecules were given an initial velocity of 100 Å/ps, which is about 10,000 m/s. The velocity changes will relate to the viscous friction with water molecules as well as the effect of the graphene defect. In this light, the interpretation of the 'kinetic energy' change shown in Fig. 5d and the conclusions drawn based on the simulations seems rather ambiguous. The description of the simulations in the methods section is too brief; it would be better to provide more detail in the SI.

We address the concerns of Reviewer 1# as follows:

Comments:

This is an interesting manuscript describing PCP through defects in single layer graphene. The experiments are of good quality and generally support the model and conclusions. A few questions should be answered:

Response:

We greatly appreciate the reviewer for the positive comments. Following the reviewer's suggestions, additional experiments have been performed and point-by-point responses to all the specific questions have been provided below.

Question 1:

What is the ionic strength in the experiments with water?

Response:

To address the reviewer's question, the Debye and Huckel formula is used to estimate the ionic strength: $I = \frac{1}{2} \sum_{i=1}^n C_i Z_i^2$, where n represents the number of ions in solution, i represent the specific ion in solution. c_i is the molar concentration of ion i (M, mol/L), z_i is the charge number of that ion, and the sum is taken over all ions in the solution. The \sum represents the summation of concentrations and valences of all ions.

In our experiments, 1 g SPMA (4.06 mmol) and 1 mL METAC (4.0 mmol) were dissolved in water-methanol mixture (1:0.5 mL), respectively, for the polymerization. Thus, the ionic strength can be calculated as follow:

$$I_{SPMA} = \frac{1}{2} \times [c(1+)^2 + c(1-)^2] = 2.70 \text{ mol/L}; \text{ in the same way, } I_{METAC} = 2.66 \text{ mol/L.}$$

However, it should be noted that the ionic strength of the solution will increase continuously during the proceeds of polymerization, since Cu species slowly dissolve from Cu plate in this process (*Polym. Chem.*, 2015, **6**, 2726-2733). Thus, the actual ionic strength should be dynamic and a bit higher than above calculated value. The calculation of the exact ionic strength in the reaction is challenge. However, if we believe that the dissociation of Cu ions from Cu plate is exclusively induced by ligand (PMDTA, 18.4 μ L, 0.088 mmol) and each dissolved Cu ions (e.g. Cu^{2+}) have to coordinate with a ligand, then the maximum ionic strength can be calculated as follow:

$$I_{SPMA} = \frac{1}{2} \times [c(1+)^2 + c(1-)^2 + c(2+)^2 + c(1-)^2] = \frac{1}{2} \times [2.70 + 2.70 + 0.058(2+)^2 + 0.058(2-)^2] = 2.93 \text{ mol/L}; I_{SPMA} = 2.89 \text{ mol/L.}$$

As such, the range of ionic strength of each polymerization solutions can be given: $2.70 \text{ mol/L} < I_{SPMA} > 2.93 \text{ mol/L}$, and $2.66 \text{ mol/L} < I_{METAC} > 2.89 \text{ mol/L}$.

Question 2:

How are the water and counter ions treated in the DFT-B simulations?

Response:

In the present work, we have considered neutral simulation boxes with monomers composed by positive (e.g. METAC) and negative (SPMA) radicals depending on the monomers under study. The inclusion of counter-ions is not so necessary in our approach, since there is no net charge in the system, but only electrically polarized molecules, e.g. in the charged monomers the charge is mostly localized on the N(+) or S(-) sites. In all simulations we used explicit water with the necessary set of parameters taken from the DFTB+ code, in which the interactions among water molecules are given by dipole-dipole as well as van der Waals interaction.

Question 3:

The OH-passivated defects are mentioned, as far as this reviewer can tell, for the first time at the bottom of page 8 - are other defect terminations important as well (viz. Fig. S12 of ref. 10)? The brief text on the MD simulation details seems to indicate thought has been given to this topic - perhaps the authors can expand on that thought? Would differences in termination open the possibility to gate the process somehow?

Response:

The reviewer's comment is highly constructive and appreciated. In the previous version of manuscript, we used four types of hydroxyl (OH) and methyl group (CH₃) terminations as representative graphene edges to get insight into the interaction of graphene defects with monomers during translocation. On account of the reviewer's suggestion, in this revision, we have carried out additional calculations of other terminations (e.g. pyrylium-like ether terminations, *Nat. Commun.* 6, 6539, 2015) for graphene defect edges as well as concentrations of these terminations (e.g., OH, CH₃ and ether) in order to gain further insight into the translocation process (see Fig. R1).

Figure R1 | Top and side views of the optimized graphene defects with different terminations (OH, CH₃, and ether passivations). The dimension of the pore only allows us to consider twelve functional groups for each passivation state. Hence, for a 50% of OH concentration, the pore will be passivated with six OH groups and six CH₃ or ether groups.

The new results have been included in the Supplementary Figs. 24.

We found that in all cases the translocation process for the neutral monomer (MMA) is much faster and easier than charged ones, as evidenced by the lowest values of translocation time τ and the loss of kinetic energy ΔE_K (Fig. R2). In addition, the translocation of MMA is nearly independent to the OH concentration at graphene edges. In contrast, the translocation of charged monomers through graphene defects is obviously affected by the concentrations of OH edges. This is due to the stronger interaction between charged monomers (METAC and SPMA) and the graphene edge groups, which also enlarges the fluctuations of graphene defects dimension (Fig. R3). These results are in well agreement with our experimental results. An additional factor controlling the translocation time, besides the OH-monomer interaction, is the strength of the OH-water dipole interaction, which leads to a gradual increase (dynamically, i.e. along the MD simulation) in the number of water molecules around the pore. The OH-water dipole interaction at graphene defect edges firstly leads to a faster translocation process of

METAC and SPMA at low OH concentration, which turns slower when OH concentration is dominant. This is a result of the more enhanced interactions between charged monomers and OH groups. The change of kinetic energy before and after the translocation process displays a similar trend as a function of the OH concentration (Fig. R2b), in which SPMA shows the highest energy loss for all cases. The results are continuous with ether terminated graphene defects, as shown in Fig. R4. In this case, both values of τ and ΔE_K decrease after increasing the ether concentration, due to the hole diameters getting larger (Fig. R3), hence, the monomers have less interaction with graphene edge groups during the translocation process.

Figure R2 | Simulation of monomers translocation through graphene defects. Variation of the (a) translocation time τ and (b) the change of kinetic energy E_K as a function of the OH concentration for the monomers in the initially CH_3 -passivated defect. $E_K = E_{K,\text{bef}} - E_{K,\text{aft}}$, with $E_{K,\text{bef}}$ and $E_{K,\text{aft}}$ as the kinetic energy of the monomer before and after the translocation process, respectively.

The new results have been included in the main text Figs. 5d and e.

Figure R3 / Variation of the pore size D_H as a function of the X (X = OH or ether) termination concentration for MMA (left panel), METAC (central panel), and SPMA (right panel). The bars at each value represent the range where the pore size fluctuates during the simulation.

The new results have been included in Supplementary Fig. 26.

Figure R4 / Variation of the (a) translocation time τ and (b) the change of kinetic energy ΔE_K as a function of the ether concentration for the monomers in the initially OH-passivated defect. $\Delta E_K = E_{K,bef} - E_{K,af}$, with $E_{K,bef}$ and $E_{K,af}$ as the kinetic energy of the monomer before and after the translocation process, respectively.

The new results have been included in Supplementary Fig. 27.

Comments:

As "very" is not a quantifiable term, its numerous appearances in the main text should be avoided. Overall, an intriguing finding that should be reviewed again before acceptance.

Response:

We thank the reviewer for the constructive comment. In the revised manuscript, the word of "very" is excluded. We hope that our revisions have adequately addressed the reviewer's concerns.

We address the concerns of Reviewer 2# as follows:**Comment 1:**

I was a pleasure to read this manuscript. It is well written, thoughtful and poses an interesting research question. The authors have also been very rigorous in the design and execution of the experiments that they report in this paper. That being said, I have a number of thoughts about the question the authors are asking as well as the insight that this study provides and how much of interest and importance the work is for the broader field.

Response:

We greatly appreciate for the reviewer's constructive comment on the work. The substantial concerns that reviewer raised have been carefully addressed as follows.

Comment 2:

The question that motivates the work in this paper is whether molecules can pass through a single-layer of graphene. This is also reflected in the title, which mentions "translocation through graphene" as well as in the conclusions, where the authors write that "we found that monomers can diffuse through a single layer of graphene". However, what the data in the paper really show (and which the

authors also mention in the discussion) is that monomer transport does not take place strictly through graphene, but instead takes place through defects / grain boundaries. These are two very different things! Transport does not take place through graphene, but is related to the defects and microstructure of the CVD deposited graphene film. As a consequence, this paper does not teach the reader about graphene, but rather about the effects of defects!

Response:

We thank the reviewer for the very thoughtful comment. We would like to address it carefully from the following points:

Firstly, for the usage of "translocation through graphene", we agree with the reviewer that this statement is not really accurate. A better way would be "translocation through graphene via native defects", and they are used in the revised manuscript to replace the inaccurate description. But we still would like to keep the current title, since it has already 14 words. Anyway, we particularly noted this in the abstract and main text of the revision.

Secondly, CVD is considered to be one of the most promising and popular method to produce high quality monolayer graphene for future electronics and membranes. And so far almost all CVD graphenes are inevitably accompanied by native defects/grain boundaries generated during synthesis. The CVD graphene in our experiment is commercial product (Graphene Supermarket, USA), which is one of the best CVD graphene (large area monolayer, $I_D/I_G < 0.1$, Fig. 2d and Supplementary Fig. 5) we could find out in market (proven technique). As such, the results and conclusions obtained in this work are general to CVD graphene, which are also in parallel to the abundant reported papers of CVD graphene.

Frankly speaking, we are not sure whether the monomers are still able to passing through if a perfect graphene was applied. However, we have shown that the defects concentration of graphene can be significantly increased from $I_D/I_G < 0.1$ to ca. 0.22 by the translocation of monomers (neutral) in SI-CRP. In this revision, we measured selected area electron diffraction pattern (SAED), which also shows the same indication of defect concentration increase after monomer translocation (Fig. R5b-d), since the SAED intensity of translocated graphene was thoroughly decreased in comparison pristine

graphene. Therefore, the possibility for monomers translocation through perfect graphene to generate new defects is not excluded yet, which however needs further more sophisticated experiments.

Anyway, we believe the drilling of nanoscale defects/holes on CVD graphene of this work by real molecules is of interest to future fundamental studies of molecule translocation as well as technologic applications of functional 2D membranes.

Figure R5 | The selected area electron diffraction pattern (SAED) taken from (a) pristine graphene, and (b-d) the graphene after translocation/polymerization of MMA. Please note that: (b-d) were taken from different positions of the same sample. The generation of ring-like SAED pattern is probably because of the distortion of graphene lattice since the translocation/polymerization is preferable occurred at native defects. Scale bars: 2 1/nm.

The new results have been used to replace previous Supplementary Fig. 8.

Comment 3:

The authors also propose that the findings of their work could be put to use for chemical cutting of 2D materials. If the observations that are reported in this paper, however, are not due to the chemical structure and properties of graphene, but are due to defects, I wonder how one can take advantage of this in a predictable manner.

Response:

We thank the reviewer for the thoughtful comment. We would like to address the comment as follow:

1) The CVD graphene we have used is high quality commercial product with mature technique (Ref. to the response of Comment 2), and we did not specially produce any defects on the graphene before the experiments.

2) Even though the graphene has defects and the translocation started from these defects, the “cutting” of graphene into microstructures using anionic monomer (SPMA) is strictly along initiator patterns, which means the whole graphene (including defects, boundaries and perfect lattice) on the top of initiator was degraded/removed by the monomer translocation. In repeated experiments using CVD graphene from different batches, we did not observe the fails of this “cutting” method.

3) Perfect monolayer can be obtained from mechanical exfoliation. However, the lateral size of resulting monolayer graphene is normally in a few micrometres, which is too small for our study, since the graphene flakes are easily to be detached when polymer brush was grown underneath.

4) We are sorry that the translocation through perfect graphene cannot be tested so far. But it will be very exciting if the translocation of monomer through perfect graphene under external driven force was experimentally proofed (*Nature* 467, 190, 2010; *Nature* 516, 227, 2014), and we has been keeping to try to design new experiments for this purpose. Anyway, it is really safe to say currently that the polymerization induced chemical “cutting” method is general to monolayer graphene from CVD method.

Comments:

In summary, I think this is a great set of experiments which are worth publication, but I think the work is more suited for a more specialized nano/graphene oriented journal.

Response:

We greatly thank the reviewer for the positive comment on the experiment. We hope that our revisions address the reviewer’s concerns adequately to publish on Nature Communication.

We address the concerns of Reviewer 3# as follows:

Comments:

This manuscript presents very interesting observations of surface polymerization in a system consisting of a surface-grafted pattern of initiators, coated with a single layer of CVD graphene. Different kinds of monomers (neutral, charged, different molecular weights) are introduced on the opposite side of the graphene, and the ensuing polymerization is examined by AFM and optical microscopy. The materials are also thoroughly characterized by a number of other techniques, including Raman, XPS, HRTEM. The authors find that polymer brushes grow despite of the presence of the graphene coating. Raman spectroscopy suggests that the polymer brushes form below the graphene, since the I_{2d}/I_g ratio increases after polymerization. Interestingly, the authors find that the polymer brush growth rate is almost the same as that in the absence of graphene for neutral monomers, but is reduced in the case of charged monomers. The authors also observe that polymerization with an anionic monomer 'cuts' the graphene, such that the polymer brush is no longer covered by graphene. To understand the observations, MD simulations are performed. The authors conclude that neutral molecules pass unimpeded through graphene, and that cutting of graphene is due to the interaction of the charged monomers with defects in graphene.

Overall, the experimental work is thorough and very well-executed. To the reviewer, it seems that the key conceptual novelty of this work is in the performance of a surface-initiated polymerization reaction on a graphene-coated surface, showing that it is possible to polymerize molecules in patterns below a graphene-coated surface. The work has some significant drawbacks, though. First, the manuscript is written in a highly speculative fashion, and makes several assertions that seem quite erroneous or not supported by the data. Second, the manuscript does not reflect much understanding of mass transport, but proceeds to make either incorrect or somewhat inaccurate statements about it, as seen in specific comments below. Third, the manuscript has errors and lacks clarity in some places, as outlined below. The reviewer believes that, if these issues are adequately addressed, the manuscript could be of suitable quality to be publishable in Nature Communications.

Response:

We appreciate the reviewer for the positive and very detailed comments. We also thank the reviewer for pointing out the drawbacks to deepen the study and in improve the quality of the manuscript. Following the reviewer's suggestions, the text of the manuscript has modified and the speculative words and inaccurate statements have been replaced to our best in the revision. In addition, additional experiments have been performed and point-by-point responses to all the specific comments raised have been provided below.

Questions 1:

There is a significant amount of literature on defects in CVD graphene and mass transport across single-layer CVD graphene that is directly pertinent to this work (e.g., DOI: 10.1016/j.carbon.2015.12.089). This literature is in the context of graphene synthesis/characterization, corrosion resistance imparted by graphene coatings (e.g., DOI: 10.1021/jacs.5b08729), single-layer graphene membranes, and some other papers, e.g., graphene has been shown to be a barrier for self-assembly of SAMs (DOI: 10.1021/acsnano.5b03936). This literature needs to be reviewed and the manuscript needs to be revised accordingly. For example, the abstract states that 'no results for larger molecules (than protons, ions, water) through pristine graphene were reported', but that is not quite true (e.g., DOI: 10.1002/adma.201605896).

Response:

We thank the reviewer for the constructive comment and sharing these interesting literatures. These literatures are important to us for a better understanding of our results, especially the last one. Following the reviewer's suggestion, we reviewed these literature in the introduction Page 2 line 7-11 of the revised manuscript, and the statement in abstract "*no results for larger molecules (than protons, ions, water) through pristine graphene*" was replaced by "*results for larger molecules through graphene are rare*".

Questions 1:

Page 2, line 4 from top: It is stated that literature suggests that CVD graphene is impermeable. However, there are several papers on transport of molecules across CVD graphene. While there is consensus that pristine graphene is impermeable to gases, molecules, and ions (except protons), it is also widely acknowledged in the literature that CVD graphene has defects that allow these to pass through.

Response:

We thank the reviewer for the constructive comment. We agree with the reviewer that there are many papers show that the CVD graphene of native defects could allow some gases, ions, and molecules to pass through. In this manuscript, we would like to emphasize that, under external driven force, the CVD graphene is permeable to the molecules that are much larger than graphene native defects, which resulted in the significant increase of graphene defects evidenced by Raman mapping (e.g. Fig. 3e) and SAED results (Ref. to the response to comment 2 of Reviewer 2#). To the best of the author's knowledge, the most of other reported works on the translocation through monolayer graphene is induced by concentration gradient and under pressure, and no evidence of the increase of graphene defects. We believe that the usage of real large molecules to generate additional defects on graphene under external driven force is potentially interest to produce nanoscale porous graphene for functional membranes.

Following the comment, the statement in Page 2 line 5 of the main text "*Even graphene prepared by chemical vapor deposition (CVD), which is expected to have Stone-Wales defects, is thought to be impermeable to helium under ambient conditions*" has been modified by "*Even graphene prepared by chemical vapor deposition (CVD), which is expected to have Stone-Wales defects, is theoretically demonstrated to be impermeable to helium under ambient conditions (Appl. Phys. Lett. 2008, 93, 193107)*" in the revision.

Questions 2:

Page 3 and SI Fig. 3: It is suggested that graphene grains are visible in the AFM image in SI Fig. 3. However, they are much more likely to be wrinkles than grains.

Response:

We thank the reviewer for the constructive comment. We fully agree with the reviewer that the morphology at the AFM image of SI Fig. 3 is more likely the wrinkles of graphene, generated probably by the grains of polycrystalline Cu foil surface. As such, the related statement (Page 4 line 1) is modified in the revised manuscript.

Questions 3:

Page 4, top: Acronyms need to be explained. The manuscript also uses too many acronyms that makes it hard to read.

Response:

We thank the reviewer for the constructive comment. Following the suggestion, we have carefully check the whole text to reduce the usage of acronyms, and explain all acronyms in the first use.

Questions 4:

Page 4, line 3 from top: It is stated that the Cu plate was removed, but the Cu plate has not been mentioned before.

Response:

We are sorry for the missing statement. The related text (in Page 4 line 1) has been modified in the revision as "*The graphene-covered initiator-SAM was then faced to a Cu plate as catalyst source in a*

distance of ca. 0.5 mm. The gap between initiator-SAM and Cu plate was filled by CRP solution of dimethyl sulfoxide or water"

Questions 5:

Page 4, bottom line: The sentence ‘Apparently, passage of a large molecule such as MMA through a single-layer graphene can be realized by a graphene-proximal reaction such as CRP’ is somewhat misleading. The reaction is not necessary for transport of the molecules across graphene; it merely facilitates transport by capturing any molecules that cross over and in effect establishing a concentration gradient across graphene.

Response:

We thank the reviewer for the constructive comment. In order to avoid the misleading that reviewer proposed, we removed the sentence “*Apparently, passage of a large molecule such as MMA through a single-layer graphene can be realized by a graphene-proximal reaction such as CRP*” in the revision.

Questions 6:

Page 5: ‘This suggests that the passage of MMA through graphene greatly enlarged native defects’ – There are several reasons why the I_{2d}/I_G ratio could decrease, including contamination, change of support, or polymerization stress-induced damage to graphene. There is no indication to suggest that native defects are being enlarged.

Response:

We thank the reviewer for the comment. We would like to note that the "This" in the sentence that reviewer mentioned means I_D/I_G intensity ratio, rather than I_{2d}/I_G . Due to potential misleading, we modified the sentence in the revision as "*Firstly, the I_D/I_G intensity ratio (i.e. defects concentration) increased from ca. 0.09 to ca. 0.22 (Fig. 3e), which suggests that the passage of MMA through the graphene greatly enlarged native defects of the lattice, since the size of graphene boundary defects (ca. 0.2 nm) are normally much smaller than that of MMA*". It is known that the I_D/I_G intensity ratio of

Raman spectrum is widely used to represent the defects concentrations of graphene (*Nano Lett.*, 2012, 12, 3925–3930; *Nano Lett.*, 2011, 11, 3190–3196). In addition, the new SAED results also support the Raman data that graphene defects were significantly increased after monomer translocation (Ref. to the response to the Comment 2 of Reviewer 2#).

Questions 7:

Page 5, line 4 from bottom: For 2-layer graphene, AFM imaging should be performed to conclude that there is no polymerization. SI Fig. 13 only shows an optical image.

Response:

We thank the reviewer for the constructive comment. Following the reviewer’s suggestions, AFM imaging was performed on the double-layer graphene based sample. We intentionally selected a position which has both double-layer graphene coated as well as graphene-free regions (Fig. R6). From both topographic and phase images, we can confirm that the PMMA growth on the double-layer graphene covered region was obviously obstructed, which is in clear contrast to the graphene free area (the crack area).

Figure R6 | (a) AFM topographic and (b) phase images of double-layer graphene coated BiBB-SiO₂ after the SI-CRP of MMA, identical to Supplementary Fig. 13. The hexagon network shows BiBB pattern after UV etching. Scale bar, 30 μ m.

The new results have been included in the Supplementary Figs. 15.

Questions 8:

Page 6 and top of page 10, 'chemical cutting of graphene': It seems to the reviewer that the most plausible mechanism for 'chemical cutting' of graphene is the reduced adhesion to the anionic polymer brush that forms below graphene. Since the patterns are at the 10 micron scale, the graphene will delaminate, generating the pattern. The reason why neutral or cationic monomers do not 'cut' the graphene may be due to the stronger adhesion with graphene (hydrophobic, pi-pi, and cation-defect). It is not clear to the reviewer how the conclusion about chemical cutting can be drawn from the simulations.

Response:

We thank the reviewer for this comment. If we understood it correctly, the reviewer worried that the "chemical cutting" may be caused by the mechanic tear force due to the rise of polymer brush underlying graphene. We would like to address the reviewer's concern from the following points:

1) Both PMMA (neutral) and PMETAC (cationic) brush delaminate the graphene, as evidenced by the significant rise of I_{2D}/I_G intensity ratio (from 2.8 to 7.0 in the case of PMMA, Fig. 3f). Although, the change of I_{2D}/I_G could be induced by contaminants and doping, in our experiment the I_{2D}/I_G of the graphene on the initiator-free region (region 1, Fig. 3a) is identical to pristine graphene (Fig. 3f). And the graphene of initiator-free region experienced same reaction conditions (solvent, monomer, catalyst) to that of polymer brush region did (Fig. 3a and 3f). Therefore, we can conclude that the increase of I_{2D}/I_G is due to the delamination and mechanical stretching (*Nano Lett.* 2009, **9**, 346-352; *Nano Lett.* 2009, **9**, 3100-3105) of graphene by the rise of PMMA brush underneath. The cationic PMETAC has similar delamination effect on graphene.

2) Previous experiments showed that a single layer graphene can be stretched by 20% without breaking. The polymer brush layer is about 43 -121 nm (for SPMA, Supplementary Fig. 19), and one hexangular pattern is ca. 40 μm . The applied stretching on the graphene lattice by the rise of a 121 nm (maximum) polymer brush is much less than 1%. In addition, in our experiment, the thickness of PMMA (neutral) brush and PMETAC (cationic) brush are ca. 80 nm and 110 nm, respectively. Thus,

the mechanical stretching forced on graphene induced by PMMA and PMETAC brushes are large than PSPMA brush. However, in both cases, we did not observe any "cutting" effect.

3) Additional, we stopped the reaction (i.e. SI-CRP) of SPMA in early stage (within 10 mins), to see if the graphene could still be "cut" by a very thin layer PSPMA brush. As the results shown in Fig. R7, the "cutting" of graphene layer was already occurred, although the thickness of PSPMA brush layer is only 8-10 nm, which has neglectable stretching force on graphene lattice. Therefore, we can conclude that the "cutting" of graphene by SPMA is due to the chemical interactions between SPMA and graphene, rather mechanical stretching/tearing force.

Figure R7 | Two selected positions of graphene morphologies and PSPMA brush thickness in 10 mins polymerization/translocation. (a) AFM topographic and (b) phase images, (c) corresponding height profile of the red line in (a). (d) AFM topographic and (e) phase images, (f) corresponding height profile of the red line in (d). Scale bars, 30 μm.

The new results have been included in the Supplementary Fig. 20.

4) Regarding to the reviewer's concerns that stronger adhesion of neutral or cationic monomers with graphene (hydrophobic, pi-pi, and cation-defect) that may prevent the "cutting" of graphene. In the manuscript, we have tried many types of neutral (methyl methacrylate, MMA; tert-butyl methacrylate, tBMA; oligo(ethylene glycol) methyl ether methacrylate, OEGMA₄₇₅), and cationic monomers

(methacryloyloxy)ethyl trimethylammonium chloride, METAC; 4-vinylpyridine, 4VP). In the revision, we additionally tested styrene (S), N-isopropylacrylamide (NIPAM) and 2-(dimethylamino)ethyl methacrylate (DMAEMA), as shown in Fig. R8. With such quantity of monomers of hydrophilic (e.g. OEGMA, NIPAM, etc.), hydrophobic (MMA, tBMA, etc.), strong pi-conjugation (S) and lack of pi-conjugation (e.g. MMA), we did not observe any “cutting” behaviour. As we stated in the manuscript, the “cation-defect” interaction is exactly the reason for the strong obstruction effect observed in the translocation of cationic monomers (e.g. METAC, Supplementary Fig. 22). And in contrast, the “anion-defect” interaction is the reason for the chemical “cutting” of graphene by SPMA. Quantum molecular dynamic simulations suggest that the highest loss of energy is found for anion SPMA (than METAC and MMA), and in parallel the largest degree of distortion of graphene edge was found for SPMA as well (Fig. 5), which implies the “anion-defect” interaction is the strongest interaction that leads the “cutting” effect.

Figure R8 | The optical images of polymer brushes under graphene after translocation of three different monomers: (a) styrene (S), (b) N-isopropylacrylamide (NIPAM) and (c) 2-(dimethylamino)ethyl methacrylate (DMAEMA). Scale bars: 40 μm.

Questions 9:

Page 8-9, simulations: It is not clear how the simulations can be interpreted. The molecules were given an initial velocity of 100 Å/ps, which is about 10,000 m/s. The velocity changes will relate to the viscous friction with water molecules as well as the effect of the graphene defect. In this light, the interpretation of the ‘kinetic energy’ change shown in Fig. 5d and the conclusions drawn based on the

simulations seems rather ambiguous. The description of the simulations in the methods section is too brief; it would be better to provide more detail in the SI.

Response:

We thank the reviewer for the constructive comment. The reviewer is right about the influence of the viscous friction with the water molecules on the kinetic energy E_K of the monomers during the translocation process. To carefully address the reviewer's concern, we performed additional simulations of different terminations of graphene defect edges to compare the energy changes before and after translocation, and the results can be seen in the Fig. R9.

Figure R9 | Kinetic energy E_K versus simulation time for fully (a) OH-passivated and (b) CH₃-passivated graphene defects.

The new results have been included in the Supplementary Figs. 25.

The translocation energy kinetics vs. time evolution are investigated on the three monomers (MMA, METAC and SPMA) through fully OH- (left panel) and CH₃- (right panel) passivated defects by MD simulation. The filled circles represent in all cases the initial (at shorter times) and final times of the translocation process through the pore. The relevant region to better understand the influence of water-induced friction is the time window between 0 (arbitrarily scaled initial simulation time) and the beginning of the translocation process (filled circles), i.e. the time window where the monomers do not feel the influence of graphene nanopore. As we can see in both panels, but most clearly for the case of the neutral monomer (blue solid lines), the kinetic energy reduction in this region is

considerably smaller than that of after the translocation process (e.g. ~ 1.5 eV vs. ~ 13.5 eV for the neutral case).

For the charged monomers, we also performed additional simulations to increase the previously mentioned relevant time window by increasing the distance of the monomers respect to the graphene surface up to ~ 30 Å, initially considered ~ 10 Å. In this case, the larger monomers are able to move for a longer time in the water environment before starting the translocation process and a similar behaviour to the neutral case was observed. Indeed, the kinetic energy reduction due to the interaction with water molecules were ~ 2.5 eV and ~ 1 eV for METAC and SPMA, respectively.

Therefore, these results clearly indicate that the change in kinetic energy arising from the water-mediated friction process is as a rule much smaller than the changes related to the translocation process. This is the reason why we only associate the strong change in kinetic energy with the translocation process through the graphene nanopores.

Reviewers' comments:

Reviewer #1 (Remarks to the Author):

The reviewer thanks the authors for addressing all of the suggested changes, which is appreciated. This reviewer recommends the work for publication.

Reviewer #2 (Remarks to the Author):

First of all, I would like to applaud the authors for the time and care that they have taken to respond to the comments by the different reviewers.

I also appreciate their feedback on my own comments and I am glad that we agree that it is most likely transport through defects and not through graphene that the authors are reporting. This is indeed appropriately corrected in the main text. Should this article eventually be published, then I would insist however that the title also be corrected accordingly. The authors refer to a word limit as the reason why they have not done this, but I really think that in a final published version (in whichever journal that may be) the title must be changed. Leaving the title in its current form is just misleading the readers.

An open question for me is still whether the work is of sufficient importance to justify publication in Nature Communications. I would stand by my previous report in which I indicated that this paper is probably a better fit for a more nanoscience oriented journal. After re-reading the paper in its revised version, I even more think it is more suited for a more specialized journal, in particular since there still seem to be many open questions as to the precise mechanism of the translocation (which I believe just is based on defects).

Reviewer #3 (Remarks to the Author):

The authors have revised the manuscript, which addresses many of the reviewer comments. A few points, however, remain to be addressed as follows.

On page 2, the work is motivated by the questions: "Can possibly larger (than graphene intrinsic defects) monomers pass through single-layer CVD graphene? If so, what happens to graphene?" Here, from reading the introduction (e.g. on page 3, "Even graphene prepared by chemical vapor deposition (CVD), which is expected to have Stone-Wales defects, is theoretically demonstrated to be impermeable to helium under ambient conditions.⁷ As such, research has shown that graphene can be used as an effective barrier to oxidation of metal surfaces under certain conditions,^{8,9,10} as well as barrier for the deposition of self-assembled monolayers.¹¹ However, in another scenario, ions and molecules have been observed to be able to transport through single layer graphene (via intrinsic defects) promoted by external pressure or concentration gradient.^{12,13,14}") it appears to imply that defects are Stone-Wales. However, ref. 14 reports several nanometer-sized defects in CVD graphene. So the question "Can possibly larger (than graphene intrinsic defects) monomers pass through single-layer CVD graphene" seems to imply that the defect size is known, and that the molecules tested are larger than the defect size. However, no detailed characterization of defect size is provided. Therefore, the question seems to be ill-posed and needs to be revised.

Regarding the mechanism of chemical "cutting", the new experiments show that it is not related to mechanical stress due to polymerization. However, given that the graphene is gone even when the polymerized layer is only 8-10 nm in size supports the hypothesis that the mechanism of chemical "cutting" is the loss of adhesion between the graphene and the underlying substrate. On page 10, it is stated "the observed chemical "cutting" of graphene by anionic monomers is related to their

highest loss of energy and highest degree of distortion of the graphene (sub)nanodefects found during the translocation process.” How can this conclusion be drawn from the simulations, which take place under conditions (high initial kinetic energy >10 eV range) that are very different than those in the experiments (room temp ~25 meV)? Furthermore, there is no clear causal link between the passage of monomer through graphene and change in the I_d/I_g ratio, as several other things are progressing in parallel. For example, is it possible that free radicals generated during the simultaneously ongoing polymerization reaction are creating new defects in graphene? The reviewer does not see how the conclusion that the patterning of graphene is caused by chemical “cutting” that is related to somehow tearing of graphene as monomers pass through defects is supported by data. As such, the mechanism remains speculative. Either more conclusive evidence needs to be provided, or the manuscript should not claim that the mechanism is identified.

We address the concerns of Reviewer 1# as follows:

Comment:

The reviewer thanks the authors for addressing all of the suggested changes, which is appreciated. This reviewer recommends the work for publication.

Response:

We thank the reviewer for the recommendation to accept the paper for publication. We agree that the suggested corrections by the reviewer significantly improved the manuscript and allowed us to better describe the importance of our scientific findings.

We address the concerns of Reviewer 2# as follows:

Comment 1:

First of all, I would like to applaud the authors for the time and care that they have taken to respond to the comments by the different reviewers. I also appreciate their feedback on my own comments and I am glad that we agree that it is most likely transport through defects and not through graphene that the authors are reporting. This is indeed appropriately corrected in the main text. Should this article eventually be published, then I would insist however that the title also be corrected accordingly. The authors refer to a word limit as the reason why they have not done this, but I really think that in a final published version (in whichever journal that may be) the title must be changed. Leaving the title in its current form is just misleading the readers.

Response:

We thank the reviewer for the positive evaluation of our improved manuscript. It really required a lot of work and we agree with the reviewer that it was worth the effort.

Following the reviewer's suggestion, we replaced the "Single-Layer Graphene" in the title with "Single-Layer CVD graphene", which is exactly what we used for the study. And there is common consensus that single layer CVD graphene always comes with defects and grain boundaries. As such, the title has been

modified to "*Polymerization-Driven Monomer Passage through Single-Layer CVD Graphene: From Unimpeded Translocation to "Chemical Cutting" of Graphene*". Actually, we have considered other terms such as "Single-Layer Defective Graphene" and "Single-Layer Graphene Defects", but this would also not reflect the finding we made, since readers would be misled and think that we had created defects on graphene before the translocation.

Comment 2:

An open question for me is still whether the work is of sufficient importance to justify publication in *Nature Communications*. I would stand by my previous report in which I indicated that this paper is probably a better fit for a more nanoscience oriented journal. After re-reading the paper in its revised version, I even more think it is more suited for a more specialized journal, in particular since there still seem to be many open questions as to the precise mechanism of the translocation (which I believe just is based on defects).

Response:

After our first, very surprising results on the very high mass translocation through graphene and the minimal reduction of the reaction rate surface polymerization under the graphene, we immediately thought of *Nature Communications* as the ideal platform to report such very fundamental findings. The more results we got the more we were convinced about the importance of our findings for a broad readership and not only for the "nanoscience community". As such, three reviewers have commented that this work is very interesting, of high novelty and the experiments generally support the conclusions. This is also the reason why we put so much effort to get this work to be published in *Nature Communications* and not in a more specialized journal. The substantial concern on the mechanism of the translocation process has been carefully addressed according to the comments of Reviewer #3 in this revision. We hope that the reviewer will follow our enthusiasm on the reported work, because it describes fundamental phenomena observed on a very popular and widely used nanomaterial.

By reviewing the other works on molecule translocation through graphene, we believe that we are also in line with the common practice and understanding of defects in graphene, see for instance:

Ref. 1: Golovchenko, J. A, et al. Graphene as a subnanometre trans-electrode membrane. *Nature* **467**, 190-193 (2010).

Ref. 2: Achtyl, J. L., et al. Aqueous proton transfer across single-layer graphene. *Nat. Commun.* **6**, 6539 (2015).

Ref. 3: Karnik, R. Selective Nanoscale Mass Transport across Atomically Thin Single Crystalline Graphene Membranes. *Adv. Mater.* **29**, 1605896 (2017).

We address the concerns of Reviewer 3# as follows:

Comment 1:

The authors have revised the manuscript, which addresses many of the reviewer comments. A few points, however, remain to be addressed as follows.

On page 2, the work is motivated by the questions: “Can possibly larger (than graphene intrinsic defects) monomers pass through single-layer CVD graphene? If so, what happens to graphene?” Here, from reading the introduction (e.g. on page 3, “Even graphene prepared by chemical vapor deposition (CVD), which is expected to have Stone-Wales defects, is theoretically demonstrated to be impermeable to helium under ambient conditions.⁷ As such, research has shown that graphene can be used as an effective barrier to oxidation of metal surfaces under certain conditions,^{8,9,10} as well as barrier for the deposition of self-assembled monolayers.¹¹ However, in another scenario, ions and molecules have been observed to be able to transport through single layer graphene (via intrinsic defects) promoted by external pressure or concentration gradient.^{12,13,14}”) it appears to imply that defects are Stone-Wales. However, ref. 14 reports several nanometer-sized defects in CVD graphene. So the question “Can possibly larger (than graphene intrinsic defects) monomers pass through single-layer CVD graphene” seems to imply that the defect size is known, and that the molecules tested are larger than the defect size. However, no detailed characterization of defect size is provided. Therefore, the question seems to be ill-posed and needs to be revised.

Response:

We greatly thank the reviewer for the positive comment on our revised manuscript.

We agree the reviewer's comment on the question on Page 2. We are sorry for this negligence and thank the reviewer for pointing this out. If we speculate on the defect size and the size of the translocated molecules, we should have measured the defect density and size of graphene. As this is very challenging to perform on the scale of our experiments, we have omitted the phrase "*...possibly larger (than graphene intrinsic defects)...*" and rephrased the question to: "*Can possibly also organic molecules such as monomers pass through single-layer CVD graphene? If so, what happens to graphene? Is the size and/or the charge of the transported monomers of importance?*"

Comment 2:

Regarding the mechanism of chemical "cutting", the new experiments show that it is not related to mechanical stress due to polymerization. However, given that the graphene is gone even when the polymerized layer is only 8-10 nm in size supports the hypothesis that the mechanism of chemical "cutting" is the loss of adhesion between the graphene and the underlying substrate. On page 10, it is stated "the observed chemical "cutting" of graphene by anionic monomers is related to their highest loss of energy and highest degree of distortion of the graphene (sub)nanodefects found during the translocation process." How can this conclusion be drawn from the simulations, which take place under conditions (high initial kinetic energy >10 eV range) that are very different than those in the experiments (room temp ~25 meV)? Furthermore, there is no clear causal link between the passage of monomer through graphene and change in the I_d/I_g ratio, as several other things are progressing in parallel. For example, is it possible that free radicals generated during the simultaneously ongoing polymerization reaction are creating new defects in graphene? The reviewer does not see how the conclusion that the patterning of graphene is caused by chemical "cutting" that is related to somehow tearing of graphene as monomers pass through defects is supported by data. As such, the mechanism remains speculative. Either more conclusive evidence needs to be provided, or the manuscript should not claim that the mechanism is identified.

Response:

After a further careful discussion and comparison of experimental and modelling data, we agree with the reviewer that our strict statement can not withhold the fact that other/additional reason(s) are possible for the observed "chemical cutting" of graphene. We admit that it's really a complicated and challenge work to precisely simulate and observe the "chemical cutting" process by computer modelling. We also agree that the precise mechanism for the monomer translocation is still an open question, and need to be further studied experimentally and theoretically. Therefore, following the reviewer's second suggestion, we have revised the related sentence in the manuscript to state the discussion of the mechanism in a moderate way.

Changes made (Page 10, first paragraph):

Old version:

The simulation results clearly support the conclusions drawn from experiments: i) a strong obstruction effect observed in both cationic and anionic monomers can be explained by the higher loss of kinetic energy found during the translocation process through the graphene pore; ii) the observed chemical "cutting" of graphene by anionic monomers is related to their highest loss of energy and highest degree of distortion of the graphene (sub)nanodefects found during the translocation process.

New version:

The combined experimental and simulation results indicate that electrostatic interactions between charged species (monomers) and the functionalized graphene nanodefects are dominating the translocation process. Moreover, the much higher energy loss experienced by the charged monomers may explain i) the strong obstruction effect observed in the translocation of both cationic and anionic monomers, and ii) the observed chemical "cutting" (in the case of anionic monomers), since the larger dissipated energy into graphene may lead to a larger degree of structural distortion of the nanodefects.

To address the reviewer's concern that radicals or other factor in parallel with a radical polymerization would cause the change of Id/Ig ratio, we placed single layer graphene on bare SiO₂/Si into a radical polymerization solution, in which the initiator was solubilized in the solution. In this case, all of the

initiators, monomers and radicals can closely contact the graphene surface, but at one side of the graphene (thus no translocation). After 24 h polymerization, we could detect high yield of polymers (PMMA > 70% and PSPMA > 90%, respectively) in the solution, but there is no change on the I_d/I_g ratio of graphene.